# Low-Rank Matrix Recovery from Noise via an MDL Framework-Based Atomic Norm

**DOI:** 10.3390/s20216111

**Published:** 2020-10-27

**Authors:** Anyong Qin, Lina Xian, Yongliang Yang, Taiping Zhang, Yuan Yan Tang

**Affiliations:** 1School of Communication and Information Engineering, Chongqing University of Posts and Telecommunications, Chongqing 400065, China; 2School of Computer Science and Technology, Chongqing University of Posts and Telecommunications, Chongqing 400065, China; 2017211059@stu.cqupt.edu.cn (L.X.); 2018211426@stu.cqupt.edu.cn (Y.Y.); 3College of Computer Science, Chongqing University, Chongqing 400030, China; tpzhang@cqu.edu.cn; 4Faculty of Science and Technology, University of Macau, Macau 999078, China; yytang@umac.mo

**Keywords:** atomic norm, low-rank matrix recovery, minimum description length principle, robust principal components analysis

## Abstract

The recovery of the underlying low-rank structure of clean data corrupted with sparse noise/outliers is attracting increasing interest. However, in many low-level vision problems, the exact target rank of the underlying structure and the particular locations and values of the sparse outliers are not known. Thus, the conventional methods cannot separate the low-rank and sparse components completely, especially in the case of gross outliers or deficient observations. Therefore, in this study, we employ the minimum description length (MDL) principle and atomic norm for low-rank matrix recovery to overcome these limitations. First, we employ the atomic norm to find all the candidate atoms of low-rank and sparse terms, and then we minimize the description length of the model in order to select the appropriate atoms of low-rank and the sparse matrices, respectively. Our experimental analyses show that the proposed approach can obtain a higher success rate than the state-of-the-art methods, even when the number of observations is limited or the corruption ratio is high. Experimental results utilizing synthetic data and real sensing applications (high dynamic range imaging, background modeling, removing noise and shadows) demonstrate the effectiveness, robustness and efficiency of the proposed method.

## 1. Introduction

Low-rank matrix recovery is important in many fields, such as image processing and computer vision [1,2,3], pattern recognition and machine learning [4,5,6] and many other applications [7,8,9]. Due to the sensor or environmental reasons, the observations used in these fields are readily corrupted by noise or outliers, and so the given data matrix *Y* can be decomposed into low-rank and sparse components.

Principal components analysis (PCA) [10] has been used widely to search for the best approximation of the underlying structure (unknown low-rank matrix *X*) of the given data. In addition, stable performance can be obtained via singular value decomposition (SVD) when the data are corrupted only by a small amount of noise. Due to the presence of gross outliers in modern applications, the robust variant of PCA—called robust PCA (RPCA)—has also been used to reject outliers [11,12]:(1)$$\underset{X,E}{\mathrm{argmin}}rank\left(X\right)+\gamma {∥E∥}_{0},\;\;s.t.\;Y=X+E,$$
where the parameter γ>0 is a regularization parameter, rank(X) denotes the rank of matrix $X\in R^{m\times n}$ (rank(X)=r) and ${∥E∥}_{0}$ is the number of non-zero entries in the sparse matrix *E*. Unfortunately, solving Equation (Equation 1) is an NP-hard problem. Instead, Candés et al. [12] solved an approximated problem by convex optimization under rather weak assumptions:(2)$$\underset{X,E}{\mathrm{argmin}}{∥X∥}_{*}+\gamma {∥E∥}_{1},\;\;s.t.\;Y=X+E,$$
where ${∥X∥}_{*}=\sum_{i}\sigma_{i}\left(X\right)$ is the nuclear norm of *X* ($\sigma_{i}\left(X\right)$ denotes the *i*-th singular value of *X*) and ${∥E∥}_{1}$ represents the $l_{1}$-*norm* of the sparse matrix *E*. Various approaches can be used to solve Equation (Equation 2) effectively [13,14].

Wright et al. [11] and Candés et al. [12] proved that the performance of Equation (Equation 2) will approach stability only by using more observations (larger *n*). However, the number of observations (*n*) is typically limited in many image processing and computer vision problems due to physical constraints. Moreover, when the number (*n*) is very limited, we note that existing methods based on Equation (Equation 2) do not reject some outliers well, such as moving objects in surveillance video [15,16,17], shadows in face images [12], and saturations in low dynamic range (LDR) images [2,18,19].

It is well known that the rank (*r*) of *X* and γ both influence the final results of RPCA decomposition. Unfortunately, the target rank and regularizing parameter γ are uncertain in Equations (Equation 1) and (Equation 2), where conventional approaches need to tune the rank of *X* and γ to achieve the desired goal. However, $\gamma =1/\sqrt{max\{m,n\}}$, which is set by the typical approaches, is not the best value [15]. The goal of model selection is to select the most appropriate model from the set of candidates; for example, selecting the appropriate parameters. The information theoretic criteria (ITC) is often used to solve the model selection problems by minimizing a penalized likelihood function via a specific criterion, such as the Akaike information criterion (AIC) and the Bayesian information criterion (BIC). The minimum description length (MDL) principle is also motivated by an information theoretic perspective and can avoid assumptions regarding prior distribution [20,21]. Instead, Ramirez et al. [15,22] used the minimum description length (MDL) principle [23] to avoid estimating the parameter γ. The MDL principle selects the best low-rank approximation from RPCA decomposition sequences, which are obtained via different values of γ. Liu et al. [17] employed structured sparse decomposition to solve the regularizing parameter issue in RPCA, where they replaced the static parameter γ by adaptive settings for image regions with distinct properties in each frame. However, an accurate rank is crucial for recovering the low-rank matrix and rejecting the outliers completely. An example of a scene is shown in Figure 1. The RPCA fails to recover the low-rank matrix and capture the illumination sudden changes.

Atoms are the fundamental basis of the representation of a signal. The atomic norm hull is the set of the fundamental elements. Moreover, the atomic norm induced by the convex hull of all unit-norm one-sparse vectors is the $l_{1}$-*norm*, and the nuclear norm is induced by taking the convex hull of an atomic set, in which the elements are all unit rank matrices [24,25,26]. To address issues such as the limited number of observations, the rank of *X* and the regularizing parameter γ, we propose a low-rank model based on the MDL principle within the devised atomic norm (MDLAN), which is also an expanded version of our published conference paper [27]. In our proposed method, we minimize the description length to select the optimum atomic sets for the low-rank matrix (*X*) and structured sparse matrix (*E*), respectively. In contrast to [15], we use the MDL principle to determine the number of atoms in the low-rank matrix, thereby avoiding tuning the rank of low-rank matrix *X*, and we also recover the sparse matrix *E* via the MDL principle. Experimental analyses show that our method can obtain a better approximation of the underlying structure of the given data when the number of observed samples is limited or if the samples have gross outliers. Thus, the proposed framework provides a nonparametric, robust low-rank matrix recovery algorithm.

The main contributions of this study are summarized as follows:(1)We present an MDL principle-based atomic norm method for low-rank matrix recovery. Unlike other model selection algorithms, the proposed MDLAN uses the description length as a cost function to select the two smallest sets of atoms that can span the low-rank matrix and sparse matrix, respectively.(2)We empirically test the MDL framework-based atomic norm and find that it outperforms the state-of-the-art methods when the number of observations is limited or if the observations have gross outliers.(3)It is difficult to address the original optimization problem for MDLAN due to the combination of description length and the atomic norm. Thus, we devise a new alternating direction method of multipliers (ADMM)-based algorithm that considers an approximation of the original non-convex problem.

The remainder of this paper is organized as follows. Section 2 briefly reviews some related research works. In Section 3, we describe the proposed MDLAN method. Section 4 presents the experimental results based on the synthetic and real datasets. Finally, we give our conclusions in Section 5.

## 2. Related Works

In the following, we briefly review recent advances in RPCA and discuss its applications in image processing and computer vision. To exactly recover *X*, some studies have replaced the rank (·) with the nuclear norm and have also replaced the number of nonzero entries with the $l_{1}$-norm, as shown in Equation (Equation 2). Candés et al. [12] proved that the rank minimization problem can be solved using Equation (Equation 1), and it also can be solved in a tractable manner by the convex relaxation version of Equation (Equation 2). They also proved that the unique solution of Equation (Equation 2) corresponded exactly to the solution of the original NP-hard problem in Equation (Equation 1) under suitable conditions.

Recently, the improvements to RPCA have been generally divided into two categories. One category focuses on the structured sparse component *E* in Equation (Equation 2) [28,29]. For example, Xin et al. [30] replaced $l_{1}$-*norm* with an adaptive version of generalized fused lasso (GFL) regularization [31], which takes into account the spatial neighborhood information of the foregrounds in a video sequence.
(3)$$\underset{X,E}{\mathrm{argmin}}{∥X∥}_{*}+\gamma {∥E∥}_{gfl},\;\;s.t.\;Y=X+E,$$
where the generalized fused lasso ${∥E∥}_{gfl}$ can be viewed as a combination of two common regularizers; i.e., the $l_{1}$-*norm* and the total variation (TV) penalty [32].
$${∥E∥}_{gfl}=\sum_{l=1}^{n}\{∥e^{\left(l\right)}{∥}_{1}+\lambda_{1}\sum_{(i,j)\in N}w_{ij}^{\left(l\right)}\left|f_{i}^{\left(l\right)}-f_{j}^{\left(l\right)}\right|\}$$
where $e^{\left(l\right)}$ is the *l*-th column of the sparse matrix *E*, N is the spatial neighborhood set, $\lambda_{1}$ is a tuning parameter and $w_{ij}=exp\left(\frac{-∥y_{i}^{\left(l\right)}-y_{j}^{\left(l\right)}{∥}_{2}^{2}}{2\sigma^{2}}\right)$ (σ≥0 is a tuning parameter (empirically set). Ebadi et al. [33] dynamically estimated the support of the sparse matrix *E* via a superpixel generation step [34] to impose the spatial coherence onto the structured sparse outliers. Shah et al. [35] replaced the $l_{1}$-*norm* in Equation (Equation 2) with hybrid $l_{1}/l_{2}$-*norm*, which can promote the spatial smoothness in the support set of the structured sparse outliers.

Another category focuses on the low-rank component *X* in Equation (Equation 2) [36]. For example, Cabral et al. [37] and Guo et al. [38,39] replaced the *X* with UV, and the relationship ${∥X∥}_{*}=\min_{U,V}\frac{1}{2}{∥U∥}_{F}^{2}+\frac{1}{2}{∥V∥}_{F}^{2}$ holds, where $U\in R^{m\times r}$, $V\in R^{r\times n}$ and ${∥\cdot ∥}_{F}$ represents the Frobenius norm. In addition, Guo et al. [38,39] employed an entropy term to restrict the support of the outliers. Hu et al. [40] proposed an approximation of the target rank by the truncated nuclear norm, which only minimized the smallest min(m,n)−r singular values. T-H Oh et al. [2] proposed the minimization of the partial sum of the singular values instead of minimizing the nuclear norm. Thus, the formulation of the partial sum can be written as follows:(4)$$\underset{X,E}{\mathrm{argmin}}\left|rank(X)-r\right|+\gamma {∥E∥}_{1},\;\;s.t.\;Y=X+E$$

The rank minimization algorithms for RPCA have inspired many applications in image processing and computer vision, such as image alignment [41], background subtraction [12,17], high dynamic range (HDR) imaging [2,42] and image restoration [43,44]. However, the clean data are easily corrupted by gross noise/outliers, or the amount of given data can be limited by factors related to the sensor or human error [2,11,45]. The available methods based on RPCA have difficulty solving these problems. In the present study, we propose an algorithm based on MDL and the atomic norm to overcome these difficulties; i.e., an unknown target rank *r*, the regularizing parameter λ and deficient observations or gross outliers.

## 3. An MDL Principle-Based Atomic Norm for Low-Rank Matrix Recovery

In this section, we will present the concept of the atomic norm and the MDL principle, respectively. We also provide the unified form of the low-rank model, which is based on the atomic norm. We then propose the new low-rank matrix recovery method (MDLAN) based on the MDL principle and atomic norm, as well as the optimization algorithm.

### 3.1. Atomic Norm

First, we provide a definition of an atomic norm and some assumptions regarding the set of atoms (A). We also assume that the set A is origin-symmetric (i.e., A∈A if and only if −A∈A). The atomic norm [24] is the gauge function induced by A:
$${∥X∥}_{A}:=\underset{t>0}{\inf}\left\{t:X\in t\cdot conv\left(A\right)\right\}$$
where conv(A) denotes the convex hull of A. In fact, the atomic norm is changed into many familiar norms when specifying the atomic set. The dual norm of ${∥\cdot ∥}_{A}$ is defined by
$${∥X∥}_{A}^{*}:=\sup\left\{\langle X,A\rangle ,a\in A\right\}$$
where the inner product is defined as $\langle X,A\rangle =tr\left(X^{T}A\right)$ for the matrix and tr(·) denotes the trace of a matrix. The dual atomic norm is crucial for producing the atomic set in our case.

**Sparsity-inducing norm:** The sparsity-inducing atomic set can be expressed as
$$A_{S}:=\left\{\pm E_{ij}\in R^{m\times n},i=1,2,\cdots ,m,j=1,2,\cdots ,n\right\}$$
where $E_{ij}$ denotes a matrix, where the *(i,j)-th* entry of the matrix is 1 and the others are zeros. Any *k*-sparse matrix in $R^{m\times n}$ is a linear combination of *k* elements from the atomic set defined above.

**Low-rankness-inducing norm:** The low-rankness-inducing atomic set can be written as
$$A_{L}:=\{Z\in R^{m\times n}{|rank\left(Z\right)=1,∥Z∥}_{F}=1\}$$
where $Z\in R^{m\times n}$ represents a rank-1 matrix with unit Frobenius norm. For any matrix $X\in R^{m\times n}$, ${∥X∥}_{A_{L}}={∥X∥}_{*}=\sum_{i}\sigma_{i}\left(X\right)$ ($\sigma {\left(X\right)}_{i}$ denotes the *i-th* singular value of the matrix *X*).

### 3.2. Atomic Norm-Based Low-Rank Matrix Recovery

The unified form of the low-rank model (Equation (Equation 2)) based on the atomic norm can also be expressed as follows:(5)$$\underset{X,E}{\mathrm{argmin}}{∥X∥}_{A_{L}}+\lambda {∥E∥}_{A_{S}}\;\;s.t.\;Y=X+E$$
where λ is the regularizing parameter. To simplify the presentation, we define a linear operator as follows.

**Definition** **1.**
*Given a set*
$\Psi =\left\{\psi_{1},\ldots ,\psi_{\left|\Psi \right|}\right\}\subset R^{m\times n}$
*, we define a linear operator*
$F_{\Psi }:R^{\left|\Psi \right|}\to R^{m\times n}$
*by*
(6)$$F_{\Psi }\alpha =\sum_{k=1}^{\left|\Psi \right|}\alpha_{k}\psi_{k}\;\;\;\;\;\forall \alpha \in R^{\left|\Psi \right|}$$

*From Equation (Equation 6), it follows that the adjoint operator*
$F_{\Psi }^{*}:R^{m\times n}\to R^{\left|\Psi \right|}$
*is given by*
(7)$$F_{\Psi }^{*}X=\left[\langle X,\psi_{1}\rangle ,\ldots ,\langle X,\psi_{\left|\Psi \right|}\rangle \right]\;\;\;\;\;\forall X\in R^{m\times n}$$


From Definition 1, it follows that the specific forms of the atomic norm in Equation (Equation 5) are given by
(8)$${∥X∥}_{A_{L}}:=\inf\left\{\sum_{i=1}^{\left|\Psi \right|}\alpha_{i}:X=F_{\Psi }\alpha ,\;\alpha_{i}\geq 0,\Psi \subset A_{L}\right\}$$
(9)$${∥E∥}_{A_{S}}:=\inf\left\{\sum_{i=1}^{\left|\Phi \right|}\beta_{i}:E=F_{\Phi }\beta ,\;\beta_{i}\geq 0,\Phi \subset A_{S}\right\}$$
where α, β are the vectors of the scalar coefficients, and $\alpha =\{\alpha_{1},\alpha_{2},\cdots ,\alpha_{i},\cdots \}$, $\beta =\{\beta_{1},\beta_{2},\cdots ,\beta_{i},\cdots \}$.

### 3.3. Minimum Description Length Principle

The MDL principle works as an objective function that balances a measure of the goodness of fit with the model complexity and searches for a model *M* from the set of possible models, M. In the MDL framework, a model M∈M that describes the given data *Y* completely with the fewest number of bits is considered the best. The MDL problem is formulated as follows:(10)$$\hat{M}=\underset{M\in M}{\mathrm{argmin}}L\left(Y,M\right)$$
where the codelength assignment function L(Y,M) defines the theoretical codelength required to describe (Y,M) uniquely. A common implementation of the MDL framework uses the Ideal Shannon Codelength Assignment ([46], Ch.5) to define L(Y,M) in terms of a probability assignment P(Y,M) as L(Y,M)=−logP(Y,M)=−log(P(Y|M)P(M)). Thus, we obtain the MDL framework
(11)$$\hat{M}=\underset{M\in M}{\mathrm{argmin}}-logP\left(M\right)-logP\left(Y|M\right)$$
where −logP(M) represents the model complexity and −logP(Y|M) represents the measure of the goodness of fit.

### 3.4. The Proposed Method

Our family of models for expressing the low-rank matrix recovery problems are defined by $M=\{\left(X,E\right):Y\leftarrow X+E,rank\left(X\right)=r,∥E{∥}_{0}=k\}$, where *r* is the truthful rank of low-rank matrix *X* and *k* represents the truthful number of non-zero entries in the sparse matrix *E*. Using these definitions, our objective function in the MDL framework can be formulated as follows:(12)$$\begin{array}{cc} \left(\hat{X},\hat{E}\right) & =\underset{M\in M}{\mathrm{argmin}}L\left(Y,M\right)\\ \; & =\underset{X,E}{\mathrm{argmin}}L\left(X\right)+L\left(E\right)+L\left(Y-X-E\right) \end{array}$$

Combining Equation (Equation 8), Equation (Equation 9) and Equation (Equation 12) yields the following MDL-based atomic norm for low-rank matrix recovery (MDLAN) model:(13)$$\underset{\Psi ,\Phi }{\mathrm{argmin}}\sum_{i}L\left(\alpha_{i}\psi_{i}\right)+L\left(F_{\Phi }\beta \right)+L\left(Y-F_{\Psi }\alpha -F_{\Phi }\beta \right)$$

The basic idea of the proposed MDLAN is to find two smallest sets Ψ and Φ, while minimizing the $L(Y-F_{\Psi }\alpha -F_{\Phi }\beta )$. The cost function in Equation (Equation 13) is non-convex in (Ψ,Φ), and we relax it with an alternative objective function in order to effectively handle the proposed problem. The codelength of low-rank matrix *X* can be written as $\sum_{i}L\left(W\psi_{i}\right)$, where $W\in R^{m\times m}$ is in a lower triangular form. Minimizing the description length of sparse matrix ($L(F_{\Phi }\beta )$) is replaced by minimizing $\theta {∥E∥}_{1}$, where $\theta =\frac{1}{k}\sum_{i=1}^{k}\left|\beta_{i}\right|$. The encoding schemes of the low-rank matrix and sparse matrix are given in Appendix A.

### 3.5. Optimization by ADMM

As with other research works [1,2,45], in this work, the equation Y=X+E still holds. Then, the original problem (Equation (Equation 13)) can be reformed as follows:(14)$$\begin{array}{cc} \left(\hat{X},\hat{E}\right) & =\underset{X,E}{\mathrm{argmin}}\sum_{i}L\left(W\psi_{i}\right)+\theta {∥E∥}_{1}\\ \; & s.t.\;Y=X+E \end{array}$$

Here, we employ the alternating direction method of multipliers (ADMM) method [13,47] to solve this constrained optimization problem. The augmented Lagrangian function of Equation (Equation 14) is
(15)$$\begin{array}{cc} L_{\mu }\left(X,E,U\right)= & \sum_{i}L\left(W\psi_{i}\right)+\theta {∥E∥}_{1}+\langle U,Y-X-E\rangle \\ \; & +\frac{\mu }{2}{∥Y-X-E∥}_{F}^{2} \end{array}$$
where μ is a positive scalar, $U\in R^{m\times n}$ is the Lagrange multiplier and 〈·,·〉 denotes the inner product operator. The ADMM consists of the following iterations: (16)$$\begin{array}{cc} & X^{t+1}=\underset{X}{\mathrm{argmin}}L_{\mu^{t}}\left(X,E^{t},U^{t}\right) \end{array}$$(17)$$\begin{array}{cc} & E^{t+1}=\underset{E}{\mathrm{argmin}}L_{\mu^{t}}\left(X^{t+1},E,U^{t}\right) \end{array}$$(18)$$\begin{array}{cc} \; & U^{t+1}=U^{t}+\mu \left(Y-X^{t+1}-E^{t+1}\right) \end{array}$$

The two subproblems (Equations (Equation 16) and (Equation 17)) are convex optimization problems which are solved while fixing another variable. Algorithm 1 summarizes the whole recovery procedure of recovering the low-rank matrix and rejecting the sparse outliers alternately. The logic that underlies the proposed MDLAN method is that the codelength cost of adding a new atom to the model is usually very high, and so adding a new atom is only reasonable if its contribution is sufficiently high to produce the largest decrease in the other part; i.e., the constrained term Y−X−E=0.
**Algorithm 1** ADMM [13,47] for the MDLAN method.**Input:** Observation $Y\in R^{m\times n},\hat{r}=min\left\{m,n\right\}$.1:Initial $X^{1}=0\left(m,n\right)$, $E^{1}=0\left(m,n\right)$, t←1, $\mu^{1}>0$, ρ>1,$\theta^{1}=1$. 2:**repeat** 3:    //Lines4−7solveEquation(19).4:    $G^{t}=Y-E^{t}+\frac{1}{\mu^{t}}U^{t}$.5:    $\Psi ,\alpha \leftarrow \max_{\Psi \subset A_{L}}\left\{\langle G^{t},\Psi \rangle :\left|\Psi \right|\leq \hat{r}\right\}$6:    $s^{t}=\left(L\left(W\psi_{1}\right),L\left(W\psi_{2}\right),\cdots ,L\left(W\psi_{\hat{r}}\right)\right)$7:    $X^{t+1}\leftarrow F_{\Psi }\langle \alpha ,I_{\frac{1}{\mu^{t}}}\left(s^{t}\right)\rangle$8:    //Line9solvesEquation(24).9:    $E^{t+1}\leftarrow S_{\frac{\theta^{t}}{\mu^{t}}}\left[Y-X^{t+1}+\frac{1}{\mu^{t}}U^{t}\right]$  10:    $U^{t+1}\leftarrow U^{t}+\mu \left(Y-X^{t+1}-E^{t+1}\right)$  11:    $\mu^{t+1}\leftarrow \rho \mu^{t}$  12:    $\theta^{t+1}\leftarrow the\;mean\;of\;\left|E^{t+1}\right|$13:    t←t+1.14:**until** converged.
**Output:** optimal $X^{t}$, $E^{t}$


#### 3.5.1. Recovering the Low-Rank Matrix

The subproblem in Equation (Equation 16) can be formulated as follows:(19)$$\underset{X}{\mathrm{argmin}}\frac{1}{\mu^{t}}\sum_{i}L\left(W\psi_{i}\right)+\frac{1}{2}{∥X-G^{t}∥}_{F}^{2}$$
where $G^{t}=Y-E^{t}+\frac{1}{\mu^{t}}U^{t}$. When we obtain the candidate atomic set of the low-rank matrix, we only need to select the suitable atoms. Since the low-rank matrix is a combination of *r* atoms, we first determine the candidate set Ψ by the dual atomic norm.
(20)$${∥X∥}_{A_{L}}^{*}:=\sup\left\{\langle X,\psi \rangle ,\psi \in A_{L}\right\}$$

This is equivalent to finding at most $\hat{r}=min\left\{m,n\right\}$ atoms to maximize
(21)$$\underset{\Psi \subset A_{L}}{\mathrm{argmax}}\left\{\langle G^{t},\Psi \rangle :\left|\Psi \right|\leq \hat{r}\right\}$$

By the Eckart–Young theorem, the atoms Ψ are obtained from the SVD of $G^{t}$, as $\Psi ={\left\{u_{i}v_{i}^{T}\right\}}_{i=1}^{\hat{r}}$, where $u_{i}$ and $v_{i}$ are the *i*-th principal left and right singular vectors, respectively (the singular value $\alpha_{i}$ is the coefficient of the corresponding atom $\psi_{i}$ and $\alpha_{1}\geq \alpha_{2}\geq \cdots \geq \alpha_{\hat{r}}$). This result ensures that the selection atoms achieve the supremum in Equation (Equation 20) and that the optimal solution will actually lie in the set Ψ. Minimizing Equation (Equation 19) and estimating the rank of the truthful low-rank matrix indicates that the selection atoms must compromise between minimizing the codelength L and being near to $G^{t}$. We can add a new atom to the low-rank matrix *X* in proper order to move opposite to the worst possible direction of the optimization problem (Equation 19). To address this optimization problem efficiently, we propose a weighted formulation [48] of description length minimization that is designed to democratically penalize the codelength of selected atoms.
(22)$$\underset{X}{\mathrm{argmin}}\frac{1}{\mu^{t}}\sum_{i}v_{i}s_{i}+\frac{1}{2}{∥X-G^{t}∥}_{F}^{2}$$
where $v_{i}\in \left\{0,1\right\}$ denotes the *i*-th element of vector *v* and vector $s=(L\left(W\psi_{1}\right),L\left(W\psi_{2}\right),\cdots ,L\left(W\psi_{\hat{r}}\right))$. $v_{i}=1$ indicates that the atom $\psi_{i}$ is selected to add to the low-rank model and the atom $\psi_{i}$ makes a sufficiently high contribution to decrease the term ${∥X-G^{t}∥}_{F}^{2}$. $v_{i}=0$ indicates that the atom $\psi_{i}$ is not selected. Thus, the subproblem (Equation 22) has a closed-form solution by varying the shrinkage operator; i.e, $X=F_{\Psi }\langle \alpha ,I_{\frac{1}{\mu^{t}}}\left(s\right)\rangle$. Where $I_{\tau }\left(x\right)$ is a variant of the shrinkage operator, defined as
(23)$$I_{\tau }\left(x\right)=\left\{\begin{array}{c} 1,x-\tau \geq 0,\\ 0,otherwise, \end{array}\right.$$
it has been shown that the number of selection atoms is the rank of the truthful low-rank matrix, *r*.

#### 3.5.2. Rejecting the Sparse Outliers

The subproblem in Equation (Equation 17) can also be formulated as follows:(24)$$\underset{E}{\mathrm{argmin}}\frac{\theta^{t}}{\mu^{t}}{∥E∥}_{1}+\frac{1}{2}{∥E-(Y-X^{t+1}+\frac{1}{\mu^{t}}U^{t})∥}_{F}^{2}$$

To efficiently minimize the $l_{1}$-norm and the proximity term in Equation (Equation 24), the soft-thresholding (shrinkage) method is employed. We can obtain the solution of the subproblem in Equation (Equation 24) as $S_{\frac{\theta^{t}}{\mu^{t}}}\left[Y-X^{t+1}+\frac{1}{\mu^{t}}U^{t}\right]$, where $S_{\tau }\left[x\right]=sign\left(x\right)max\left(\left|x\right|-\tau ,0\right)$ is the soft-thresholding operator [49].

### 3.6. Discussion

Why does the proposed MDLAN recover the best approximation of the low-rank and sparse matrices, even though the number of observations is limited or the observations have gross outliers? In our MDL framework, the recovery of the low-rank matrix *X* by solving Equation (Equation 16) or Equation (Equation 19) is performed with the aim of finding the smallest set of atoms in $A_{L}$ that can span *X*, so it is equivalent to
(25)$$atoms\left(X\right)=\min_{\Psi }\left\{\left|\Psi \right|:\Psi \subset A_{L},X\in span\left(\Psi \right)\right\}$$
and recovering the sparse matrix by solving Equation (Equation 17) or Equation (Equation 24) is equivalent to
(26)$$atoms\left(E\right)=\min_{\Phi }\left\{\left|\Phi \right|:\Phi \subset A_{S},E\in span\left(\Phi \right)\right\}$$
where we note that rank(X)=atoms(X) and ${∥E∥}_{0}=atoms\left(E\right)$. Thus, this theory (Equations (Equation 25) and (Equation 26)) can ensure that the proposed algorithm recovers the low-rank matrix accurately and rejects the outliers.

As shown in Algorithm 1, the proposed MDLAN can find the candidate atoms for the truthful low-rank matrix and sparse outliers, respectively, and then decides which atom to add to the model according to the MDL principle. Estimating the rank of the truthful low-rank matrix *X* correctly is the key to recovering the low-rank matrix accurately, and it also contributes to rejecting all outliers. Similarly, rejecting all outliers will contribute to the search for the best approximation of the truthful low-rank matrix *X*.

## 4. Experiments

We evaluate the proposed method using both synthetic data sets and real sensing application examples to verify its effectiveness and robustness. In all experiments, we use the default parameters for the methods compared.

### 4.1. Experiments with Synthetic Data

To compare the proposed method (MDLAN) with state-of-the-art methods on synthetic data, we synthesize a ground-truth low-rank matrix $X_{0}\in R^{m\times n}$ of *rank*-*r* and a sparse matrix $E_{0}\in R^{m\times n}$ with *k* nonzero entries that simulates the bad data due to sensor malfunction. The low-rank matrix is a linear combination of *r* arbitrary orthogonal basis vectors, and the weights used to span the vector are sampled randomly from the uniform distribution U(0,5). The *k* entries from $X_{0}$ are corrupted by random noise from N(0,1). We refer to $\frac{∥X_{0}{-X∥}_{F}}{∥X_{0}{∥}_{F}}$ as the normalized root mean squared error (NRMSE).

#### 4.1.1. Comparison of the Success Ratio

We use the recoverability results to verify the robustness of RPCA and our method (MDLAN) with respect to the number of samples (*n*), synthetic data dimension (*m*) and corruption ratio (*p*). For each pair—(n,p) and (m,p)—we run 50 trials and report the overall average NRMSE of the trials. If the recovered low-rank matrix *X* has an NRMSE value smaller than ε (ε=0.01), we consider that recovery is successful. The magnitudes of the colors in Figure 2 and Figure 3 indicate the success probability. The larger red areas indicate the more robust performance of the algorithm.

Figure 2 shows the success ratio using RPCA and the proposed method with ranks 2, 4, 6 and 8. We fix m=4900 and vary *n* and *p*. When the number of observations is deficient or the corruption ratio is large, the proposed method can obtain competitive results. Both methods exhibit similar behaviors when more samples are available or the corruption ratio is small.

We also performed experiments in which we fixed n=15 and vary *m* and *p*. As shown in Figure 3, the proposed method yields more robust results than RPCA for the rank 1 and 3 cases. Figure 3 shows that the dimension (*m*) does not have a particularly significant effect on the results. However, the number of observations and corruption ratio severely affect the final recovery results.

#### 4.1.2. Comparisons with Other Low-Rank Ratrix Approximations

We also perform experimental comparisons of a rank minimum-based method (RPCA) [12], MDL principle-based method (LR-MDL) [15], conditional gradient with enhancement and truncation based on atomic norm (CoGEnT) [25], generalized fused lasso foreground modeling (BSGFL) [30], partial sum of singular values-based method (PSSV) [2], low-rank matrix recovery via robust outlier estimation (ROUTE) [39], factor group-sparse regularization for low-rank matrix recovery (FGSR) [50] and low-rank matrix recovery via subgradient method (SubGM) [51]. We verify the robustness of RPCA, LR-MDL, CoGEnT, BSGFL, PSSV, ROUTE, FGSR, SubGM and the proposed method (MDLAN) with respect to the corruption ratio. We fix m=108, n=100 (except for SubGM, where we set n=108), r=4 and vary the corruption ratio p∈[0.01,0.8]. To show more of the detail obtained by RPCA, LR-MDL, CoGEnT, PSSV, ROUTE, FGSR, SubGM and MDLAN in Figure 4, the results of BSGFL are not shown (as the spatial neighborhood information is considered in a sparse matrix, the BSGFL fails to recover the synthetic sparse matrix).

Figure 4a shows the NRMSE of the low-rank matrix for each method as a function of the corruption ratio based on the synthetic data averaged over 50 random runs. As shown in Figure 4a, when the outlier ratio is lower than 0.3, the proposed method obtains similar results to PSSV and RPCA, which are better than those produced by the other methods (LR-MDL, CoGEnT, ROUTE, FGSR and SubGM). When the outlier ratio is more than 0.3, MDLAN achieves much higher accuracy than RPCA, LR-MDL, CoGEnT, PSSV, ROUTE, FGSR and SubGM. It is clear that gross outliers exist, and thus the existing methods do not capture all the energy of the underlying structure. The results shown in Figure 4c demonstrate that only the proposed method estimates the rank of the underlying structure correctly (*rank*-4). As stated in the previous section, the proposed method finds all the candidate atoms of low-rank matrix via the atomic norm and then selects the most appropriate atoms via the MDL principle. Estimating the rank of the underlying structure correctly is crucial for recovering the low-rank matrix accurately and also benefits the outlier estimation.

The NRMSE of the sparse matrix obtained by our method in Figure 4b has smaller errors than those produced by RPCA, LR-MDL, CoGEnT, PSSV, ROUTE, FGSR and SubGM when the outlier ratio is more than 0.3. The proposed method can search for the best approximation of the sparse structure via the MDL, so it can obtain more accurate results. Moreover, compared with the other methods, the proposed approach estimates the number of nonzero entries in the sparse matrix more accurately, even when the corruption ratio is up to 0.55, as shown in Figure 4d (the number of nonzero entries recovered by LR-MDL, ROUTE and SubGM are always mn). When the corruption ratio is more than 0.55, the number of nonzero entries estimated by the proposed method is still close to the original number. To reject the outliers completely, it is necessary to recover the locations and the corresponding values of the nonzero entries accurately, which we achieved by solving Equation (Equation 24) in our MDL framework.

Table 1 shows the recovery results averaged over 50 random runs, where the corruption ratio is fixed to p= 0.05 or 0.5. When the data are corrupted with 50% outliers, the average NRMSE for the low-rank matrix using the proposed method is 0.01 and the average NRMSE for the sparse noise matrix is 0.019. In addition, MDLAN preforms better than LR-MDL, CoGEnT, BSGFL, ROUTE, FGSR and SubGM when the corruption ratio is only 0.05. In summary, the experimental results for synthetic data suggest that MDLAN performs better at recovering the low-rank matrix and rejecting the outliers from the corrupted data compared with the other state-of-the-art methods.

### 4.2. Real-World Sensing Applications

#### 4.2.1. High Dynamic Range (HDR) Imaging

Low dynamic range (LDR) images of a scene are usually captured by a sensor with different bracketing exposures. We formulate the HDR image generation problem as a rank-minimization problem, where the moving objects, noise and other nonlinear artifacts are considered as sparse outliers and our goal is to merge several LDR images into the final HDR images. We know that LDR images are linearly dependent due to the continuous camera response. Thus, we construct three observed intensity matrices $Y\in R^{m\times n}=\left[vec\left(I_{1}\right),\cdots ,vec\left(I_{n}\right)\right]$ by stacking the vectorized input images (processing each color channel individually), where *m* and *n* represent the number of pixels and images, respectively, and $I_{i}$ denotes the input image. We apply the rank minimization methods to the three corrupted matrices to separate the outliers and the background scene (low-rank term).

We apply the proposed approach to the three observed matrices $Y\in R^{699392\times 4}$ using a set of LDR images comprising four pictures taken in a forest [18]. The images contain artifacts caused by a person walking in the scene. Moreover, the wind makes the branches move, and thus there are shadows due to the wind. The final HDR results are shown in Figure 5. Compared with the results obtained by RPCA, LR-MDL, CoGEnT, BSGFL, PSSV, ROUTE and FGSR, the proposed method can recover the low-rank component (artifact-free in Figure 5g) and reject more outliers, even with only four input images (n=4). The detailed comparison in Figure 6 shows that our method can reject the outliers, such as ghosting and shadows, which are caused by the person and the wind, respectively. The reason for this is that the proposed MDLAN uses the description length as a cost function to select the two smallest sets of atoms that can span the low-rank matrix and sparse matrix, respectively. Furthermore, the proposed method, utilizing the MDL principle to select the optimal atoms, can search for the best approximation of the sparse structure. Figure 4b–d also shows that the proposed MDLAN can estimate the intensity and the number of the nonzero entries in the sparse matrix.

#### 4.2.2. Background Modeling Based on Video Sensor

We adopt the *F*-measure as the quantitative metric for the performance evaluation of the background modeling. The *F*-measure, which combines precision and recall, is calculated as follows:(27)$$F-measure=2\frac{precision\cdot recall}{precision+recall}$$
where $precision=\frac{TP}{TP+FP}$ and $recall=\frac{TP}{TP+FN}$, *TP*, *FP*, *TN* and *FN* denote the numbers of true positives, false positives, true negatives and false negatives, respectively. The higher the *F*-measure, the more accurately the outliers (foreground objects) are detected [52].

In background modeling, it is difficult to determine the correlations between video frames as well as modeling background variations and the foreground activity. It is reasonable to assume that these background variations are low-rank, while the moving objects in the foreground are large in magnitude and sparse in the spatial domain. Background estimation is complex due to the presence of foreground activity such as moving people and variations in illumination.

We first consider the example video introduced by Li et al. [53], which comprises a sequence of 1186 grayscale frames obtained from a busy shopping center. Multiple people move in the scene, and so the shadows on the ground surface vary significantly in the image sequences. To verify the effectiveness of the proposed method when the number of observations is limited, we only utilize a small number of continuous frames (*n* = 100). Each frame has resolution of 256×320, and we stack the frames as the columns in our observed matrix $Y\in R^{245760\times 100}$.

The results are displayed in Figure 7, which show that all the methods successfully detect the moving people. However, many shadows are present in the low-rank background recovered by RPCA, LR-MDL, CoGEnT, BSGFL, PSSV, ROUTE and FGSR, as shown in Figure 7b–h. By contrast, our proposed method correctly models the background scene and gives a better foreground with fewer false detections.

We then consider two sequences from the stuttgart artificial background subtraction (SABS) data set, including a “Basic” sequence and a “Clutter” sequence. The “Clutter” category of sequences contains a large number of foreground moving objects occluding a large portion of the background, which is very challenging, and we also only utilize 100 continuous frames. The results of all models on an example frame are indicated in Figure 8 and Figure 9. As shown in Figure 8, the proposed method obtains a cleaner background (no ghosting) and detects more outliers compared to the other models when the corruption ratio is high. Figure 9 demonstrates that the proposed MDLAN can recover the low-rank background (no shadow) and almost cuts the foreground correctly compared to the other models.

The average *F*-measures and running time (on a 3 GHz Core(TM) i7 CPU) of all the models on the three sequences are shown in Table 2. As illustrated in Figure 7, Figure 8 and Figure 9, the shadow is included in the sparse component, which makes the value of the *F*-measure relatively low. Table 2 indicates that the proposed method can achieve the highest *F*-measure for the three sequences and also shows better computational efficiency.

#### 4.2.3. Removing Noise and Shadows From Faces

Basri et al. [54] stated that the face recognition problem in computer vision is a low-dimensional linear model and showed that, under certain idealized circumstances, images captured by a sensor which is under variable illumination lie near an approximately nine-dimensional linear subspace known as the harmonic plane. However, due to the presence of shadows and specularities, real face images often violate the aforementioned low-rank model. It is reasonable to consider that outliers such as shadows, specularities and saturations are sparse in the spatial domain. Thus, we aimed to recover a low-rank model from the corrupted face images. The images have a resolution of 96×84, and we stack 20 face images as the columns in our observed matrix $Y\in R^{8064\times 20}$.

Figure 10a shows three images from the Extended Yale B database [55], Figure 10b–i shows the recovered low-rank components and Figure 11a–h shows the corresponding sparse components. Unlike the other methods, when the shaded area is small, MDLAN removes the shadows around the nose region (see the first and second rows in Figure 10i). When the shaded area is large, the proposed method still removes more shadows than RPCA, LR-MDL, CoGEnT, BSGFL, PSSV, ROUTE and FGSR (see the third row in Figure 10g). In addition, we add salt and pepper noise to each observed image, and the noise density is 0.2. Figure 12b–i shows the recovered low-rank components. Compared to the above methods, the proposed MDLAN can remove both noise and shadows. Thus, our technique may be useful for pre-processing training images in face recognition systems by removing such noise/outliers.

## 5. Conclusions

In this study, we introduce the MDL principle and atomic norm into the field of low-rank matrix recovery, and we propose a novel nonparametric low-rank matrix approximation method called MDLAN. The existing algorithms have difficulty tackling the proposed optimization problem; thus, we consider an approximation of the original problem. Our method selects the best atoms to search for the best approximation of the low-rank matrix, and it also can find sparse noise simultaneously. We compare the proposed approach with state-of-the-art methods using synthetic data and three real sensing low-rank applications; i.e., HDR imaging, background modeling based on a video sensor and the removal of noise and shadows from face images. The experimental results using the synthetic and real sensing data sets demonstrate the effectiveness and robustness of the proposed approach.

## Figures and Tables

**Figure 1 sensors-20-06111-f001:**
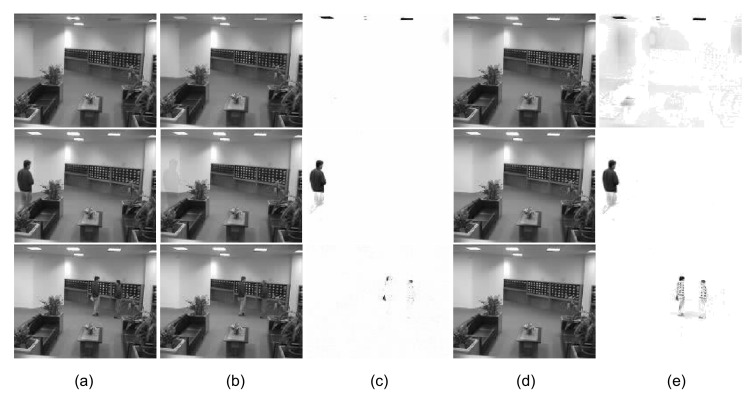
Recovered background and detection of outliers. Three frames from an 80-frame sequence taken in a lobby are presented. (**a**) Three frames from the original video *Y*, low-rank component *X* (**b**,**d**) and structured sparse component *E* (**c**,**e**) obtained by robust principal components analysis (RPCA) (**b**,**c**), and the proposed approach (**d**,**e**), respectively. The rank estimated by RPCA is 7, and so ghosting appeared in the background. By contrast, the rank estimated by our approach is 1.

**Figure 2 sensors-20-06111-f002:**
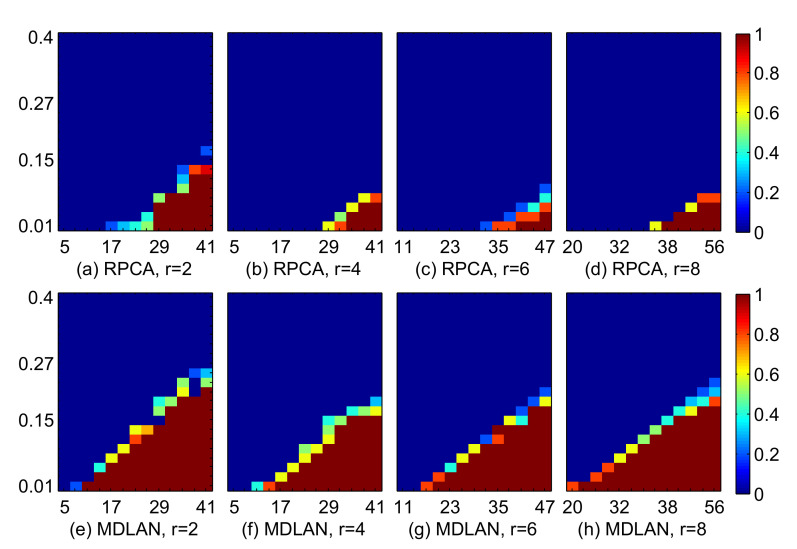
Recovery results with various numbers of observations (*n*). Comparison of robust principal components analysis (RPCA) and the proposed minimum description length atomic norm (MDLAN) method for rank 2, 4, 6 and 8 cases. The *X*-axis represents the number of samples (*n*) and the *Y*-axis represents the corruption ratio p∈[0.01,0.41]. The color magnitude represents the success ratio [0, 1].

**Figure 3 sensors-20-06111-f003:**
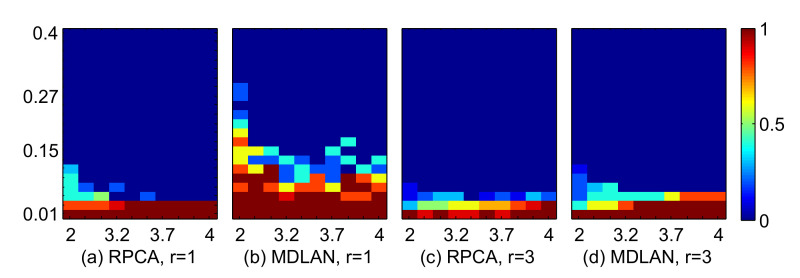
Recovery results with various numbers of dimensions (*m*). Comparison of RPCA and the proposed MDLAN method for rank 1 and 3 cases. The *X*-axis represents the log-scale row size ($log_{10}m\in \left[log_{10}100,log_{10}100,000\right]$) and the *Y*-axis represents the corruption ratio p∈[0.01,0.41]. The color magnitude represents the success ratio [0, 1].

**Figure 4 sensors-20-06111-f004:**
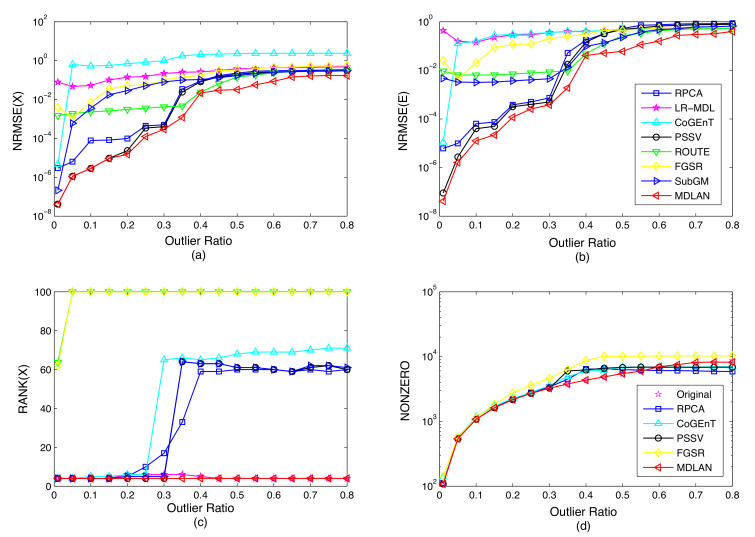
(**a**) Average normalized root mean square error (NRMSE) for the low-rank matrix, (**b**) average NRMSE for the sparse matrix, (**c**) the rank (*r*) of recovery for the low-rank matrix, (**d**) average number of nonzero entries of the recovered sparse matrix.

**Figure 5 sensors-20-06111-f005:**
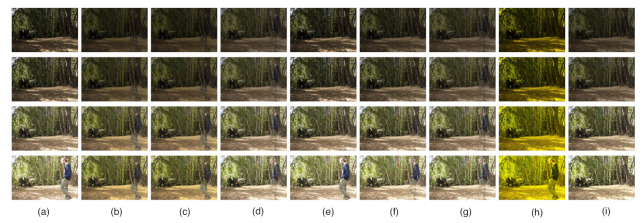
High dynamic range imaging. (**a**) Input images (four) and low-rank term *X* obtained by RPCA (**b**), LR-MDL (**c**), CoGEnT (**d**), BSGFL (**e**), PSSV (**f**), ROUTE (**g**), FGSR (**h**) and the proposed approach (**i**).

**Figure 6 sensors-20-06111-f006:**
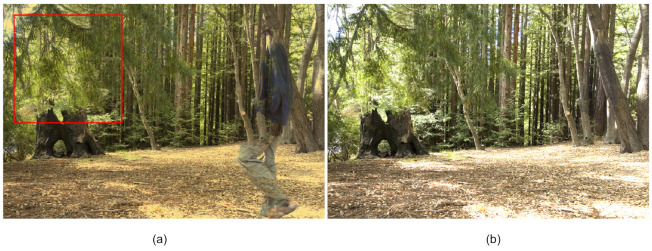
Detailed comparison of the branches and their shadows. Low-rank component obtained by RPCA (**a**) and the proposed approach (**b**).

**Figure 7 sensors-20-06111-f007:**
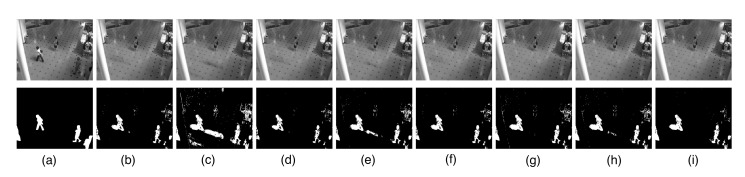
Background modeling results on the Li data set. (**a**) One frame of the original video *Y* (top) and the ground truth (bottom). Low-rank component *X* (top) and sparse component *E* (bottom) obtained by RPCA (**b**), LR-MDL (**c**), CoGEnT (**d**), BSGFL (**e**) PSSV (**f**), ROUTE (**g**), FGSR (**h**) and the proposed approach (**i**).

**Figure 8 sensors-20-06111-f008:**
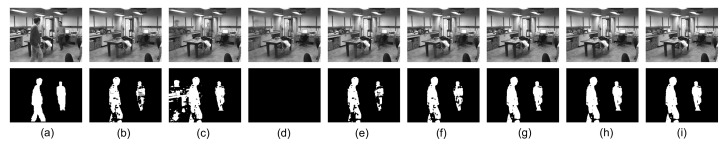
Background modeling results on the SABS data set. Each frame has a resolution of 240×320, and we stack the frames as the columns in our observed matrix $Y\in R^{76800\times 100}$. (**a**) One frame of the original video *Y* (top) and the ground truth (bottom). Low-rank component *X* (top) and sparse component *E* (bottom) obtained by RPCA (**b**), LR-MDL (**c**), CoGEnT (**d**), BSGFL (**e**) PSSV (**f**), ROUTE (**g**), FGSR (**h**) and the proposed approach (**i**).

**Figure 9 sensors-20-06111-f009:**
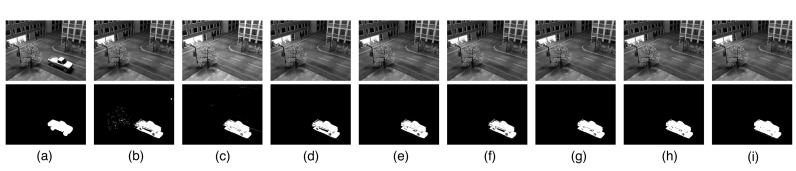
Background modeling results on the SABS data set. Each frame has a resolution of 600×800, and we stack the frames as the columns in our observed matrix $Y\in R^{480000\times 100}$. (**a**) One frames of the original video *Y* (top) and the ground truth (bottom). Low-rank component *X* (top) and sparse component *E* (bottom) obtained by RPCA (**b**), LR-MDL (**c**), CoGEnT (**d**), BSGFL (e) PSSV (**f**), ROUTE (**g**), FGSR (**h**) and the proposed approach (**i**).

**Figure 10 sensors-20-06111-f010:**
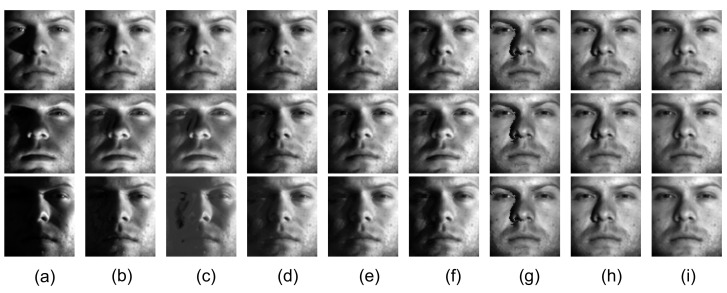
Removing shadows and specularities from face images. (**a**) Cropped and aligned images from the Extended Yale B database of a person’s face under different illumination. The size of each image is 96 × 84 pixels and 20 different illumination settings in total were used for the person. Low-rank term *X* obtained by RPCA (**b**), LR-MDL (**c**), CoGEnT (**d**), BSGFL (**e**), PSSV (**f**), ROUTE (**g**), FGSR (**h**) and the proposed approach (**i**).

**Figure 11 sensors-20-06111-f011:**
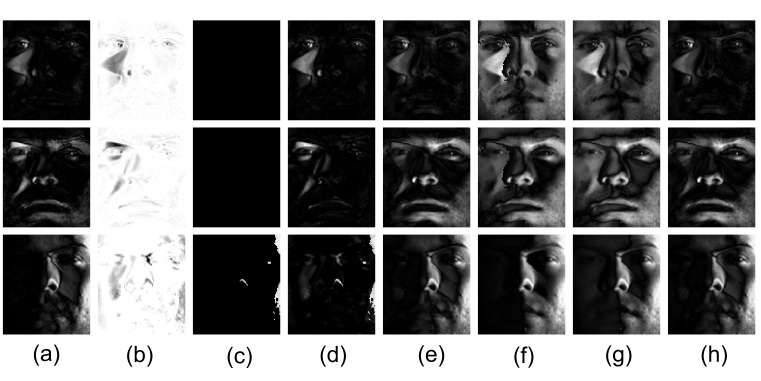
Sparse component *E* obtained by RPCA (**a**), LR-MDL (**b**), CoGEnT (**c**), BSGFL (**d**), PSSV (**e**), ROUTE (**f**), FGSR (**g**) and the proposed approach (**h**).

**Figure 12 sensors-20-06111-f012:**
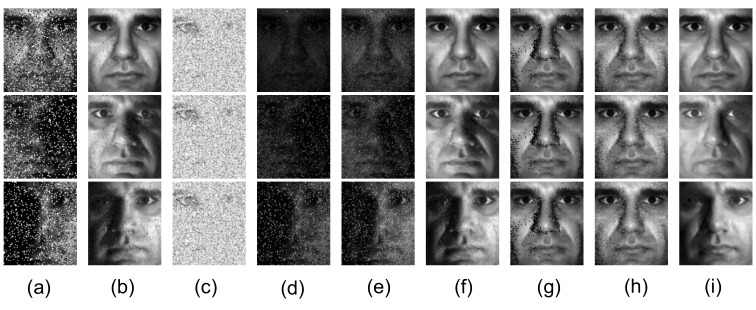
Removing noise, shadows and specularities from face images. (**a**) Cropped and aligned images from the Extended Yale B database of a person’s face under different illumination. The size of each image is 96 × 84 pixels, and 20 different illumination settings in total were used for the person. We also added salt and pepper noise to each image. Low-rank term *X* obtained by RPCA (**b**), LR-MDL (**c**), CoGEnT (**d**), BSGFL (**e**), PSSV (**f**), ROUTE (**g**), FGSR (**h**) and the proposed approach (**i**).

**Table 1 sensors-20-06111-t001:** Quantitative comparison of NRMSE for the low-rank and sparse noise matrices. LR-MDL: low-rank minimum description length; CoGEnT: conditional gradient with enhancement and truncation; BSGFL: generalized fused lasso foreground modeling; PSSV: partial sum of singular values; ROUTE: low-rank matrix recovery via robust outlier estimation; FGSR: factor group-sparse regularization; SubGM: subgradient method.

	0.05		0.5	
	**Low-Rank**	**Sparse**	**Low-Rank**	**Sparse**
RPCA	**0.000007 ± 0.0**	**0.00001 ± 0.0**	0.25 ± 0.005	0.591 ± 0.014
LR-MDL	0.061 ± 0.024	0.200 ± 0.072	0.379 ± 0.034	0.490 ± 0.024
CoGEnT	0.082 ± 0.032	1.218 ± 0.082	1.604 ± 0.585	1.541 ± 0.144
BSGFL	0.030 ± 0.004	0.275 ± 0.063	0.351 ± 0.026	10.04 ± 1.423
PSSV	**0.000002 ± 0.0**	**0.00002 ± 0.0**	0.207 ± 0.002	0.486 ± 0.005
ROUTE	0.002 ± 0.0001	0.007 ± 0.0001	0.141 ± 0.009	0.264 ± 0.015
FGSR	0.001 ± 0.0001	0.004 ± 0.0001	0.324 ± 0.007	0.454 ± 0.008
SubGM	**0.0006 ± 0.0**	0.003 ± 0.0001	0.168 ± 0.039	0.227 ± 0.005
MDLAN	**0.000002 ± 0.0**	**0.000003 ± 0.0**	**0.01 ± 0.007**	**0.019 ± 0.013**

**Table 2 sensors-20-06111-t002:** Quantitative evaluation of the background modeling, given as the F-measure and running time.

	Shopping Mall		HumanBody2		MPEG	
	**F-Measure**	**Times (s)**	**F-Measure**	**Times (s)**	**F-Measure**	**Times (s)**
RPCA	0.6975	585	0.6881	985	0.7437	4087
LR-MDL	0.5021	912	0.6294	106	0.7977	446
CoGEnT	0.6846	1534	0.0676	586	0.7735	6498
BSGFL	0.6406	11628	0.6806	2219	0.7987	14942
PSSV	0.7060	20.1	0.7377	37.1	0.7756	207
ROUTE	0.7181	62.4	0.7801	41.5	0.8110	340
FGSR	0.7175	61.8	0.7837	50.2	0.8141	304
MDLAN	**0.7536**	28.6	**0.7941**	40.7	**0.8264**	227

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
