# Peer review of "Low-Rank Matrix Recovery from Noise via an MDL Framework-Based Atomic Norm"

_sensors, 2020, doi:10.3390/s20216111_

Round 1

Reviewer 1 Report

  1. Line 259. How do you identify outliers and noises? Please explain why your method can eliminate the ghosting and shadows. 
  2. Please add quantitative analysis to the forest dataset, SABS dataset, and the face dataset. 

Reviewer 2 Report

This paper presents a low rank matrix recovery. The main limitations of this work are the following. 

  1. First the paper organization is poor confusing the riders. I suggest to the authors to include the main statement in the main article, while the rest to include in an Appendix.
  2. The section of experiments is poor. New experiments are required adding for example noise to the data
  3. What about the size of a matrix. How does the algorithm operates on very large matrices?
  4. Synthetic examples are required for understanding the overall performance the method and compared it with other SOTA approaches. 

Reviewer 3 Report

he manuscript addressed the problem of recovery of an underlying low-rank structure of clean data corrupted with sparse noise (typically outliers) in a scenario when the rank of the underlying structure is unknown. The proposed solution leverage on the MDL principle and atomic norm by finding the candidate atoms and then minimizing the orde's model. The topic is of great interest, timely, and with a number of applications. The technical part is sound, new, and well-described. The authors also put a considerable effort to provide practical examples in the area of image processing, The manuscript is easy to follow, and the core ideas are well presented. I recommend some polishing to remove some typos. My only recommendation (apart from minor polishing of the language) is to broaden a bit the literature and the introduction to include some related aspects. For example, model order selection can be performed not only by MDL. There are alternatives, some of which can be tuned to reach some predefined recovery performance, see e.g., "Model Order Selection Based on Information Theoretic Criteria: Design of the Penalty," IEEE Trans. on Signal Processing. Some insight can also be found in "Model Selection Techniques - An Overview." It could also be useful for the reader to know that there were some efforts to find bounds on the recovery capabilities in sparse settings. Although not for this specific problem, it could be interesting for the reader to find opportunities for research, in this area. For example, "Sparse recovery of noisy data using the Lasso method".
